# Transcriptomic Analysis Reveals the Role of Silver Nanoparticles in Promoting Maize Germination

**DOI:** 10.3390/plants14193022

**Published:** 2025-09-30

**Authors:** Zhipeng Yuan, Xuhui Li, Zhi Liang, Ran Li, Weiping Wang, Xiangfeng Li, Xuemei Du, Quanquan Chen, Riliang Gu, Jianhua Wang, Li Li

**Affiliations:** 1State Key Laboratory of Maize Bio-Breeding, Beijing Innovation Center for Crop Seed Technology, Ministry of Agriculture and Rural Affairs, College of Agronomy and Biotechnology, China Agricultural University, Beijing 100193, China; iuenzipo@163.com (Z.Y.); chenquanquan@cau.edu.cn (Q.C.);; 2Sanya Institute of China Agricultural University, Sanya 572000, China; 3Institute of Nanfan and Seed Industry, Guangdong Academy of Science, Guangzhou 510316, China; ylws2201@163.com

**Keywords:** maize, transcriptomic analysis, silver nanoparticles, aging tolerance

## Abstract

The germination, seedling growth, and crop productivity of maize seeds are significantly impacted by seed aging. This study investigated the efficacy of silver nanoparticles (AgNPs) as a seed priming agent for maize inbred lines exhibiting varying degrees of aging tolerance. Two inbred lines, aging-sensitive I178 and aging-tolerant X178, were used. AgNP treatment significantly promoted the germination of I178 (from 55% to 85%, compared with water treatment). Notable improvements were observed in root length, shoot length, and lateral root formation after AgNP treatment in I178. However, X178 showed no significant changes in germination and seedling growth after the AgNP treatment. Further transcriptomic analysis was performed on X178 and I178 before (water treatment) and after AgNP treatment to study genes and the expression network of the mechanism induced by AgNP promotion. In I178, AgNP treatment led to a substantial increase in differentially expressed genes (DEGs). A total of 800 DEGs were identified, with 517 being upregulated and 283 downregulated. The DEGs in I178 were mainly involved in metabolic processes, stress responses, and membrane repair. For example, genes related to lipid metabolism and membrane integrity were upregulated, along with seven genes associated with antioxidant action and redox metabolism. This indicates that AgNPs might enhance membrane stability and stress tolerance in I178. In contrast, X178 had a limited transcriptomic response to AgNP treatment. Although 874 DEGs were detected, the number of genes related to key processes like those in I178 did not change significantly, which is in line with its inherent aging tolerance. Overall, these results suggest that AgNPs can effectively improve seed vigor and counteract the negative effects of seed aging, especially in aging-sensitive maize lines. The mechanism likely occurs through regulating gene expression related to stress response, metabolic repair, and membrane stability. This study provides new insights into the molecular basis of AgNP-mediated seed vigor enhancement, which has potential implications for improving seed quality in agricultural production.

## 1. Introduction

Seed vigor is a critical determinant of successful crop establishment, particularly under suboptimal environmental conditions such as after prolonged storage [1]. Aging during seed storage, which leads to a decrease in seed quality, is a pervasive problem in agriculture that significantly impairs germination and subsequent plant growth [2]. Seed deterioration due to aging is a result of several physiological changes, including damage to cell membranes, oxidative stress [3,4,5], and the disruption of metabolic processes essential for seed germination [6]. These changes lead to a reduction in the seed’s ability to initiate successful germination, particularly when exposed to adverse environmental conditions.

In maize, a crop of significant agricultural importance, seed aging is particularly problematic for varieties with lower natural vigor, such as certain inbred lines. Aging-sensitive varieties exhibit substantial reductions in germination rates, root length, and overall seedling vigor after prolonged storage [7,8]. Several strategies, such as chemical treatments, hormonal priming, and physical processes, have been employed to mitigate the effects of seed aging, but these often come with limitations in terms of efficacy, environmental impact, and applicability to different crops [9,10]. Recently, nanotechnology, particularly the use of nanoparticles (NPs), has emerged as a promising alternative for seed priming [11,12,13,14]. Nanomaterials such as silver nanoparticles (AgNPs) are believed to enhance seed germination and vigor by facilitating nutrient uptake, modulating oxidative stress, and improving cellular membrane integrity [15,16].

Silver nanoparticles, in particular, have been shown to possess unique antimicrobial, antioxidant, and growth-promoting properties, making them suitable candidates for enhancing seed performance under stress [17]. AgNPs have been found to stimulate seedling growth by improving metabolic activity, increasing enzyme production, and enhancing membrane stability [16]. Moreover, AgNPs have been reported to enhance the stress tolerance of seeds, particularly in response to environmental stressors such as high salinity, drought, and low temperature [10]. While the effects of AgNPs on seed priming are well-documented, the molecular mechanisms through which these nanoparticles promote seed vigor, especially in aging-sensitive varieties, remain underexplored.

In this study, we investigated the effect of AgNP priming on two maize inbred lines, X178, also named CAU178 (178 of China Agricultural university), and the improved CAU178 (I178), which have similar genetic backgrounds to X178 [18], and the only difference was seed vigor or seed storage capacity. I178 is an aging-sensitive inbred line, while X178, an aging-tolerant inbred, maintained a higher level of seed vigor. We hypothesized that AgNPs would significantly improve seed germination and seedling vigor in I178, possibly by enhancing metabolic pathways and repairing aging-induced damage. In contrast, we expected that AgNPs would have less impact on the aging-tolerant inbred X178, as it is less affected by seed aging. To better understand the molecular mechanisms underlying the effects of AgNP priming, we conducted transcriptomic analysis to identify differentially expressed genes (DEGs) in both 178 inbred lines following AgNP treatment. The findings from this study provide new insights into the molecular basis of AgNP-induced seed vigor enhancement and may have significant implications for improving seed quality, especially in aging-sensitive maize varieties.

## 2. Results

### 2.1. Germination and Growth Responses to AgNP Treatment

After one month of natural storage in Jan of Sanya, the germination rates of I178 and X178 were maintained at 55% and 73%, respectively. In inbred I178, AgNP treatment (PT) resulted in a significant improvement in germination characteristics compared to the control (from 55% of the water treatment to 85% of the AgNP treatment). Similarly, root length and shoot length were significantly enhanced in the PT group, with the average root length increasing from 3.08 ± 0.12 cm to 4.71 ± 0.28 cm and the average shoot length from 1.11 ± 0.07 cm to 1.79 ± 0.11 cm. Notably, AgNP treatment also increased the lateral root count significantly. In contrast, there was no significant improvement in germination or seedling growth observed in X178 after AgNP treatment, with only a marginal increase in lateral root number (Figure 1).

### 2.2. Transcriptome Sequencing and Differential Gene Expression in Two 178 Inbreds After AgNP Treatment

To delve into the mechanism by which AgNP affects seed vigor, we conducted transcriptomic analysis on seed samples from I178 and X178 subjected to both PT and CK treatments. Principal component analysis (PCA) clearly differentiated between the various treatments and inbred lines, validating the accuracy of the sequencing outcomes (Figure 2). A total of 349,708,496 clean reads were obtained, with read counts ranging from 40,423,338 to 45,159,698 across the groups. After aligning the reads to the maize reference genome (Zm-B73-REFERENCE-GRAMENE-4.0), 98.73–99.39% (mean 99.08%) of the reads were mapped. The number of expressed genes was similar between I178 (mean 23,367) and X178 (mean 23,094) under ddH_2_O treatment, but both 178 inbreds showed an increase in the number of expressed genes following AgNP treatment (I178: mean 22,712; X178: mean 23,581) (Table 1).

These results highlight the robustness and reliability of the transcriptomic data. The high mapping efficiency (above 98%) and consistent expression profiles across biological replicates support the integrity of sequencing and sample preparation. Notably, the increase in expressed genes in both I178 and X178 after AgNP treatment suggests that nanoparticle priming may activate latent or stress-responsive transcriptional programs during seed germination. This could reflect a generalized enhancement in transcriptional activity in response to external stimuli, even in aging-tolerant genotypes. The slightly higher number of expressed genes in X178PT compared to I178PT may indicate a broader but less specialized transcriptional activation in X178, possibly due to its naturally higher vigor and more stable gene regulation network. These results reinforce the distinct transcriptomic responses between genotypes and support the downstream analysis of differentially expressed genes (DEGs) as a basis for uncovering AgNP-mediated regulatory mechanisms.

### 2.3. Identification of Differentially Expressed Genes (DEGs) Induced by AgNP Treatment

A total of 800 DEGs were identified in I178 following AgNP treatment (Log2FC ≥ 2; *q* value = 0.01), with 517 genes being upregulated and 283 downregulated, compared to the ddH_2_O-treated control (Figure 3A). In X178, 874 DEGs were identified, with 602 being upregulated and 272 downregulated.

We are aiming to study DEGs that have different expression patterns in two 178 inbreds (I178 and X178), which are standard for the genetic difference in aging sensitivity vs. aging tolerance. Also, the number of differentially expressed genes in X178 and I178 was similar. These DEGs were primarily associated with seed aging repair mechanisms and stress response pathways, suggesting that AgNPs play a role in mitigating the effects of seed aging in I178 but not in X178.

Although the total number of DEGs in X178 was slightly higher than that in I178, the nature and functional categories of these DEGs differed substantially. In I178, DEGs were more functionally concentrated in stress mitigation and membrane stabilization, while X178 DEGs were more evenly distributed across metabolic pathways, suggesting a less targeted response. Furthermore, the larger fold change range observed in I178 DEGs indicates that AgNP treatment induced more pronounced transcriptional shifts in the aging-sensitive genotype. This may imply that I178 possesses a higher degree of transcriptional plasticity when exposed to external priming stimuli, which is critical for recovering seed vigor. The presence of both upregulated and downregulated genes also suggests a complex reprogramming of gene networks rather than a simple activation, reflecting the multifactorial nature of aging recovery. These distinctions further underscore the need for a genotype-specific interpretation of nanoparticle effects on gene expression.

### 2.4. Common DEGs Induced by AgNP and Specific DEGs Different from Aging Sensitivity and Tolerant Inbreds

Among the DEGs identified, 179 genes were found to be commonly expressed in both I178 and X178 under AgNP treatment, termed common DEGs (cDEGs) (Figure 3B). A GO analysis of these cDEGs revealed enrichment in biological processes (BPs) related to cellulose biosynthesis, carbohydrate metabolism, and cell membrane repair, along with various enzyme production pathways (Figure 4). These findings suggest that AgNP treatment enhances seedling morphogenesis or seedling establishment, along with the consumption of storage substances, like carbohydrate such as starch, as well as cellular repair mechanisms and metabolism during seed germination.

Further analysis identified 141 specific differentially expressed genes (sDEGs), which were upregulated in I178 but downregulated in X178. These sDEGs were found to be involved in seed germination, metabolic processes, and stress responses. A qRT-PCR validation of nine sDEGs, including genes encoding proteins involved in germination and cellulose formation, confirmed the transcriptomic results, showing significant upregulation in I178 but not in X178 (Figure 5).

The identification of 179 common DEGs (cDEGs) suggests that AgNPs may trigger a shared basal response in both inbreds, likely related to general germination and metabolism. However, the significant number of specific DEGs (sDEGs), particularly the 141 genes showing opposite expression trends between I178 and X178, implies that the physiological status and inherent stress response of the genotype heavily influence transcriptional outcomes. These sDEGs may serve as key genetic indicators for evaluating nanoparticle responsiveness and could represent potential targets for molecular breeding. The confirmed upregulation of germination- and cell wall-related genes in I178, validated via qRT-PCR, provides strong evidence that AgNPs help override the aging-induced repression of critical growth functions. This specificity supports the hypothesis that nanopriming not only acts as a global metabolic booster but also fine-tunes particular signaling pathways in a genotype-dependent manner. These gene expression contrasts lay the foundation for functional characterization in future work.

### 2.5. Functional Pathways of sDEGs in Two 178 Inbreds After AgNP Treatment

A GO analysis of the sDEGs revealed that the upregulated genes in I178 were mainly associated with metabolic processes, stress responses, and membrane and catalytic activities (Figure 6). Additionally, the downregulated sDEGs in I178 were related to cellular processes and metabolic activities. These findings highlight the importance of membrane stability and metabolic regulation in seed vigor, particularly in an aging-sensitive inbred like I178.

The functional classification of sDEGs in I178 emphasizes the pivotal role of metabolic recovery and membrane reorganization during AgNP-induced germination enhancement. The enrichment in stress-related pathways—including those associated with oxidative stress responses, catalytic enzyme activities, and lipid biosynthesis—suggests that AgNPs help mitigate biochemical impairments accumulated during aging. On the other hand, the downregulated genes in I178 were often associated with energy-consuming growth processes, which may reflect a strategic reallocation of resources toward repair and restoration. In contrast, X178 did not exhibit strong pathway enrichment in similar categories, reinforcing the view that its basal transcriptional architecture is already optimized for aging tolerance. The contrasting functional pathways further illustrate how nanopriming interacts with the pre-existing physiological state of seeds. These findings also provide mechanistic insights into how nanopriming reshapes cellular priorities to enable rapid and resilient germination in stress-compromised seeds.

### 2.6. Membrane Repair Against Oxidation and Cell Wall Development Genes

One of the key findings of the transcriptome analysis was the identification of genes involved in plasma membrane repair and lipid metabolism. In I178, 291 genes were specifically upregulated under AgNP treatment, including genes related to lipid metabolism and membrane integrity. Notably, seven genes associated with antioxidant action and redox metabolism (Zm00001eb002510, Zm00001eb262910, Zm00001eb200160, Zm00001eb333390, Zm00001eb016980, Zm00001eb075660, and Zm00001eb283570) were significantly upregulated, and three genes associated with plant growth, plant cell wall synthesis, and development were enhanced (Zm00001eb323730, Zm00001eb267300), supporting the hypothesis that AgNPs contribute to membrane repair and enhanced stress tolerance (Figure 5). Validation tests in maize hybrids showed that the AgNP treatment had the greatest effect on the increase in lateral root number in aging-sensitive seeds (Figure 7). At the same time, the germination rate of seeds was significantly improved.

## 3. Discussion

### 3.1. Enhancement in Germination and Seedling Growth in Aging-Sensitive Maize by AgNPs

The observed selective improvement in seed performance in the aging-sensitive inbred I178 after AgNP priming reflects a broader trend reported across various crop species, where nanopriming tends to exert stronger effects on low-vigor or deteriorated seeds [19,20]. Similar genotype-dependent responses have been documented in aged rice [21], wheat [22], and watermelon [12], where nanoparticles enhanced germination and seedling vigor primarily under conditions of physiological stress. Mechanistically, AgNPs have been implicated in enhancing water uptake kinetics, modulating phytohormone signaling, and promoting the early activation of metabolic pathways related to carbohydrate mobilization [9,23]. Such effects may explain why varieties with high inherent vigor, such as X178, often show negligible benefits, a phenomenon consistent with earlier reports in seed physiology that link priming efficacy to the extent of physiological impairment [16]. In the context of maize production, the genotype specificity of nanopriming responses underscores the need for targeted application strategies to maximize cost-effectiveness and sustainability.

Building upon these findings, the genotype-specific enhancement observed in aging-sensitive seeds like I178 not only holds theoretical relevance but also offers practical implications for seed conservation and reuse. In regions with hot and humid climates, such as southern China or tropical zones, natural aging is a major threat to seed viability during storage. In these cases, AgNP-based priming may serve as a cost-effective strategy to recover germination potential, reduce seed wastage, and improve seed use efficiency. This has particular significance for long-term seed banks and commercial seed storage programs dealing with low-vigor lots. Additionally, the differential responsiveness of genotypes to AgNP treatments underscores the need for precision strategies in seed priming. Coupling nanopriming protocols with genetic background profiling could lead to customized seed treatment regimens that maximize efficacy. Interestingly, recent studies have shown that AgNPs, when co-applied with other agents such as silicon or micronutrients, produce synergistic effects on germination and early growth, suggesting the potential for multi-component priming systems in future seed technology pipelines [24].

### 3.2. Transcriptomic Insights into AgNP-Mediated Regulation of Seed Vigor

Comparative transcriptome analyses in plants primed with nanoparticles consistently highlight the modulation of genes involved in stress response, membrane repair, and primary metabolism [25]. In rice, AgNPs have been shown to upregulate ROS-scavenging enzymes and lipid metabolism genes, leading to improved membrane integrity during germination [26]. Similar transcriptomic signatures have been reported for other nanomaterials, such as cerium oxide nanoparticles in barley [27] and zinc oxide nanoparticles in chickpea [28], suggesting that enhanced stress resilience is a conserved molecular outcome of nanopriming. GO term enrichment in cellulose biosynthesis, carbohydrate metabolism, and membrane repair observed here aligns with the findings from previous studies [19,29], where structural reinforcement and metabolic activation were linked to improved germination under stress. This convergence of transcriptomic evidence across species strengthens the argument that nanopriming operates via evolutionarily conserved pathways that could be leveraged for crop improvement. In line with previous research on seed transcriptome dynamics, the observed enrichment in genes related to cellulose biosynthesis, carbohydrate metabolism, and membrane repair may reflect the activation of specific regulatory modules, such as Seed Activation Promoter Clusters (SAPCs). These clusters are known to be rapidly induced during the imbibition phase, triggering a cascade of transcriptional and metabolic events. AgNPs may potentially accelerate or amplify the activation of such clusters by modifying chromatin accessibility or through epigenetic modulation [30]. Moreover, beyond conventional gene expression changes, recent work has demonstrated that AgNPs can influence post-transcriptional regulation, including alternative splicing and non-coding RNA-mediated signaling. Such mechanisms could play a more critical role in sensitive genotypes like I178, where molecular flexibility is essential for recovery from oxidative and metabolic stress during germination [31].

### 3.3. Membrane Repair, Antioxidant Defense, and Cell Wall Development as Core Mechanisms

The integration of membrane repair, oxidative stress mitigation, and structural development emerges as a recurring mechanistic theme in nanopriming research. Transcriptomic analysis in this study revealed that seven genes associated with antioxidant activity and redox metabolism (Zm00001eb002510, Zm00001eb262910, Zm00001eb200160, Zm00001eb333390, Zm00001eb016980, Zm00001eb075660, and Zm00001eb283570) were significantly upregulated. Additionally, three genes involved in plant growth, cell wall synthesis, and development (Zm00001eb323730 and Zm00001eb267300) showed significant increases, potentially contributing to lipid metabolism and membrane repair processes. These findings support previous studies demonstrating that “AgNPs facilitate membrane repair and enhance stress resistance.”

Studies on aged soybean seeds have demonstrated that nanoparticle treatments can enhance the expression of phospholipid-transporting ATPases and lipid transfer proteins, improving membrane fluidity and stability [32]. The induction of antioxidant systems, including the ascorbate–glutathione cycle, is also widely reported in nanoparticle-primed rice [16], maize, and wheat, [33] reflecting a shared strategy to counteract ROS accumulation during early germination [10]. Cell wall remodeling genes, particularly cellulose synthase and expansin families, are often co-activated [34], supporting theories that cell wall loosening and reinforcement must occur in tandem to facilitate root and shoot emergence [35]. Cross-species comparisons suggest that these processes are not only critical for overcoming aging-induced structural damage but also for enhancing the seedling’s competitive ability under suboptimal conditions.

Notably, several of the I178-specific upregulated genes linked to membrane repair share homology with membrane-associated signaling regulators reported in Arabidopsis and rice, such as MSR2 and PLDα1. These genes are involved in calcium-dependent phospholipid signaling pathways that orchestrate membrane fusion and structural stabilization [16]. The activation of these pathways by AgNPs likely facilitates the rapid restoration of membrane integrity in aged seeds. On the antioxidant front, in addition to well-characterized enzymes like CAT, SOD, and APX, AgNP treatment may also enhance the levels of small redox molecules such as glutathione and ascorbate. These molecules help maintain cellular redox homeostasis during early germination and mitigate oxidative stress caused by seed aging [36]. Furthermore, in terms of cell wall remodeling, genes encoding members of the expansin family were also found to be significantly upregulated in I178. Expansins are critical in loosening the cell wall matrix, enabling embryo elongation and radicle protrusion through the endosperm. This structural flexibility may explain the more robust root and shoot emergence observed in AgNP-treated I178 seedlings [37].

### 3.4. Agricultural Implications and Future Perspectives

The accumulating evidence from maize and other crops suggests that AgNPs and related nanomaterials could be integrated into seed technology pipelines for targeted improvement in low-vigor seed lots [11]. However, field-scale adoption demands a rigorous assessment of long-term agronomic benefits, environmental persistence, and nanoparticle bioaccumulation risks [34]. Emerging studies highlight the potential of combining nanopriming with biostimulants, microbial inoculants, or other eco-friendly priming agents to achieve synergistic effects [17]. Furthermore, tailoring nanoparticle properties—such as size, coating, and surface chemistry—could optimize uptake efficiency and minimize toxicity, as demonstrated in recent precision agriculture research [24]. Future work should adopt integrative multi-omics approaches to map the complete signaling and metabolic networks influenced by nanopriming [38], ensuring that its application is both biologically effective and environmentally responsible.

While the experimental benefits of AgNP-based priming are evident, several agronomic and environmental variables must be considered before field-scale application. For instance, the stability and bioavailability of AgNPs can be significantly influenced by soil composition and microbial activity. Some studies have reported the rapid adsorption or oxidation of AgNPs in certain soils, leading to reduced bioactivity [39]. Thus, developing protective delivery systems—such as biodegradable coatings or encapsulated nanocarriers—could help maintain AgNP efficacy in diverse field environments. In addition, nanoparticle safety must remain a high priority. As new findings in nano-toxicology emerge, there is growing awareness about the potential genetic, epigenetic, and microbiome-level effects of nanoparticle accumulation in agricultural ecosystems [24]. Future research should therefore integrate multi-omics and long-term ecological assessments to ensure that AgNP-based priming is both biologically effective and environmentally sustainable.

## 4. Materials and Methods

### 4.1. Preparation of Silver Nanoparticles

Silver nanoparticles (AgNPs) were synthesized by the reduction of AgNO_3_ with trisodium citrate according to procedures described by Fang et al. [30] and Asta et al. [31]. Briefly, 50 mL of 1.0 mM AgNO_3_ solution was heated to boiling in an Erlenmeyer flask. Subsequently, 5 mL of 1% C_6_H_5_O_7_Na_3_ was added dropwise to this solution. During this process, the solution was mixed vigorously by the use of a magnetic stirrer. The solution was heated until a pale yellow color was observed. It was then removed from the heating surface and stirred until it cooled to room temperature.

### 4.2. Plant Materials and Seed Treatment

Two maize inbreds, I178 (aging-sensitive) and X178 (aging-tolerant), were selected for this study. New harvested seeds (January of 2025) were stored under nature conditions (95% moisture content and ~30 °C temperature) at Innovation Research and Study Valley Laboratory of China Agricultural University in Sanya Yazhou Bay Science and Technology City (109.161154 E, 18.326592 N) for one month (this is equivalent to the conditions for seeds that have naturally aged); then the seeds were primed with a 0.01 mM AgNP solution for 20 h at room temperature, while the control group was treated with deionized water (ddH_2_O) for the same duration. Unprimed seeds were also compared in the verification of the results but are not shown in the results because no significant differences were shown.

### 4.3. Germination and Seedling Measurements

Germination was assessed by placing 50 seeds on paper rolls and incubating them under standard conditions (25 °C, 8 h light/16 h dark; 3 replications). Subsequently, 10 randomly selected seedlings of each replication after 4 days of germination were collected for RNA isolation and transcriptome sequencing analysis. The germination rate, root length, shoot length, and lateral root count of rest samples (40 seeds or seedlings of each replication) were scored and measured for validation. Statistical analysis was performed using a one-way ANOVA to compare the effects of AgNP treatment between the groups.

### 4.4. Transcriptome Sequencing and Analysis

RNA was extracted from the seedlings of both the AgNP-treated and control groups, with two biological replicates per group. The RNA samples were sequenced on an Illumina NovaSeq 6000 platform, and differential gene expression analysis was conducted using the DESeq2 package with a threshold of |log_2_ fold change| ≥ 1 and an adjusted *p* value ≤ 0.05. Gene Ontology (GO) analysis was performed to classify the enriched biological processes and pathways affected by AgNP treatment.

### 4.5. RT-qPCR Validation

Similar RNA samples to those above were used for DEG validation. Total RNA was used for reverse transcription, and first-strand cDNA was synthesized using a StarScript Ⅱ RT Mix with gDNA Remover kit (GenStar, Beijing, China). RT-qPCR was carried out in triplicate for each sample using an SYBR Green Ⅰ Kit (GenStar) on a QuantStudio 6 Flex system (ABI, Waltham, MA, USA). ACTIN 1(LOC100282267) was used as an internal control (CK).

## 5. Conclusions

This study provides novel insights into the use of AgNPs as a seed priming agent for improving seed germination, particularly in aging-sensitive maize inbreds. The transcriptomic data revealed that AgNPs enhance stress tolerance and promote seed vigor through the upregulation of genes involved in membrane repair and metabolic regulation. These findings lay the groundwork for future studies on the use of nanomaterials in seed priming and their potential to mitigate the effects of seed aging in agricultural production.

## Figures and Tables

**Figure 1 plants-14-03022-f001:**
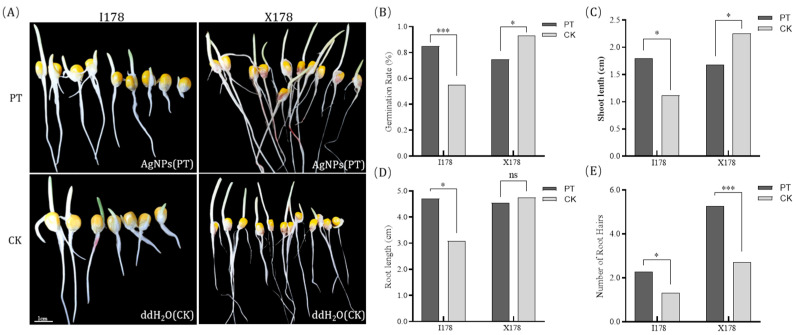
Effect of different seed priming treatments on X178 and I178. Character of 178 seeds treated by AgNPs compared with those treated with water (**A**), germination rate (**B**), shoot length (**C**), root length (**D**), and number of root hairs (**E**) of 178 seeds treated by AgNPs compared with those treated with water.* represents *p* < 0.05 (significant difference); *** represents *p* < 0.001 (highly significant difference); ns represents no difference.

**Figure 2 plants-14-03022-f002:**
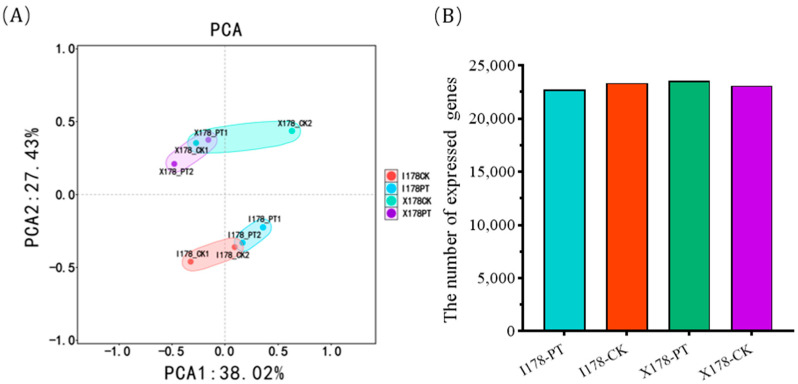
RNA-Seq of germinated 178 seeds after treatment of AgNPs. PCA of sample quality after RNA-Seq (**A**). Data quality of sequenced samples reflected by number of expressed genes detected in different samples (**B**).

**Figure 3 plants-14-03022-f003:**
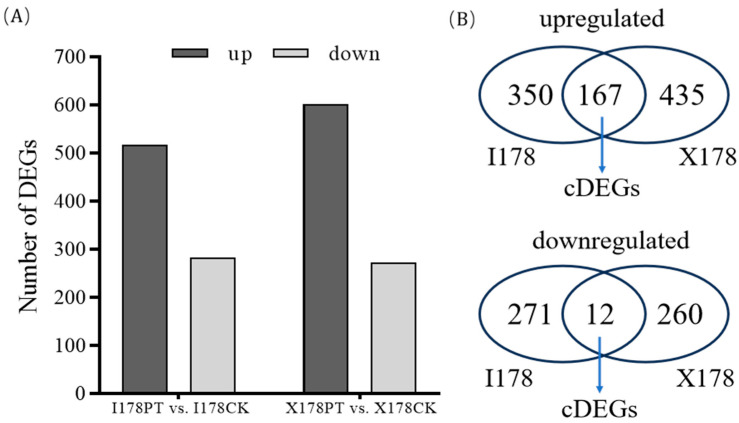
Comparative analysis of DEGs in I178 and X178 seedlings before and after AgNP treatment. (**A**) Comparison of number of upregulated and downregulated DEGs in I178PT vs. I178CK and X178PT vs. X178CK. (**B**) Venn diagram drawn using DEGs in I178 and X178, including those that are upregulated and downregulated.

**Figure 4 plants-14-03022-f004:**
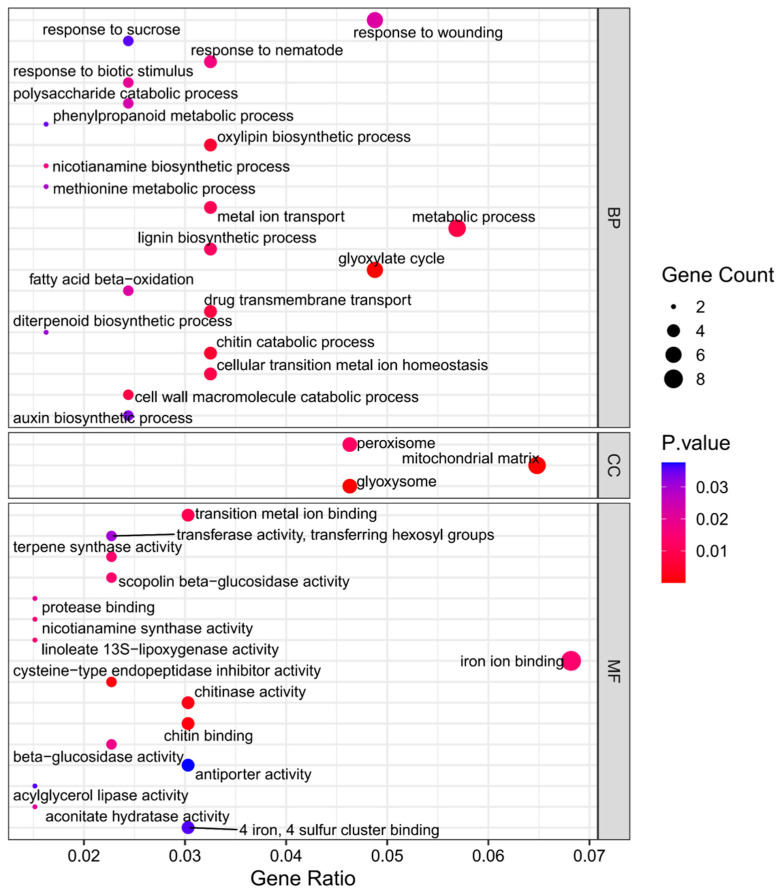
GO enrichment in AgNP-responsive cDEGs in I178 and X178 seedlings. The size and color scale of the points in the figure represent the number and significance level of DEGs in GO terms, respectively. According to the standardized classification system of gene function, it is divided into BP (biological process), CC (cellular component), and MF (molecular function).

**Figure 5 plants-14-03022-f005:**
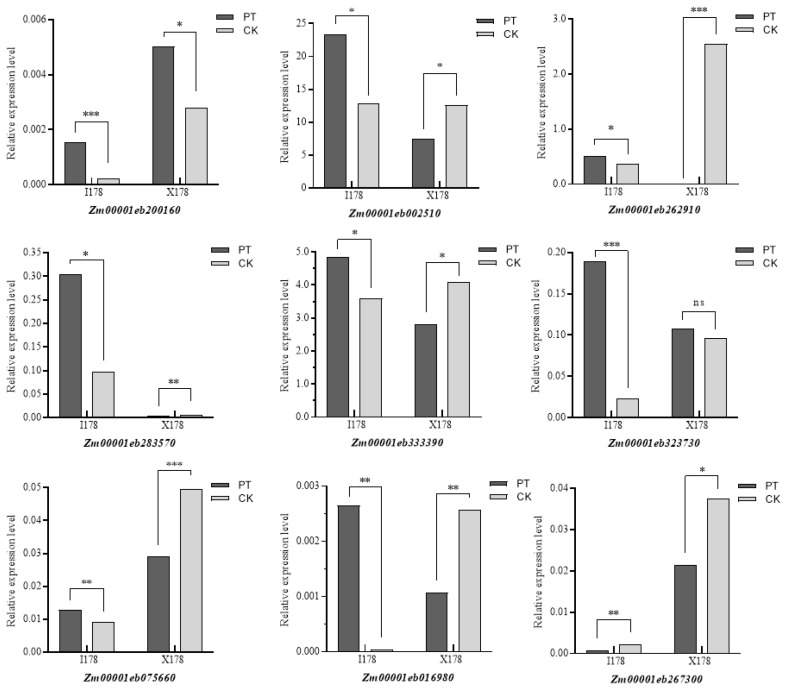
Validation of I178-specific DEGs by qRT-PCR. *ACTIN 1* was used as internal control. *p* values were calculated by one–way ANOVA; ns represents not statistically significant, * represents *p* < 0.05 (significant difference); ** represents *p* < 0.01, *** represents *p* < 0.001 (highly significant difference).

**Figure 6 plants-14-03022-f006:**
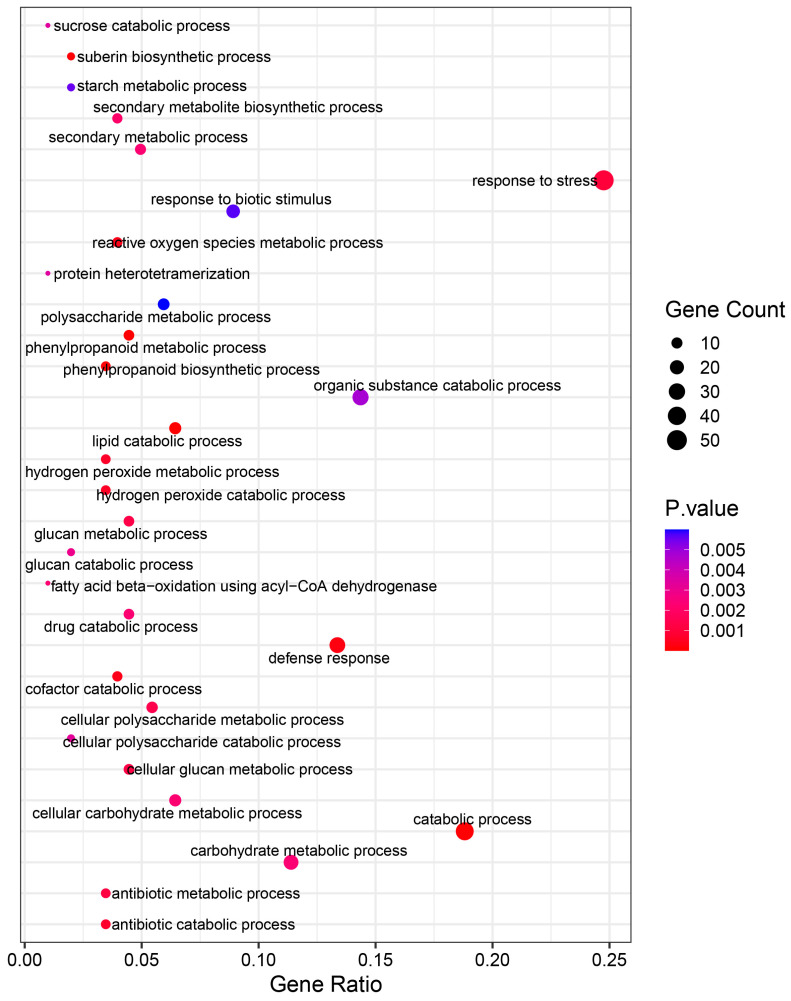
GO analysis of sDEGs which respond to AgNP treatment in I178 (I178PT vs. I178CK) seedlings. Size and color scale of points in figure represent number and significance level of DEGs in GO terms, respectively.

**Figure 7 plants-14-03022-f007:**
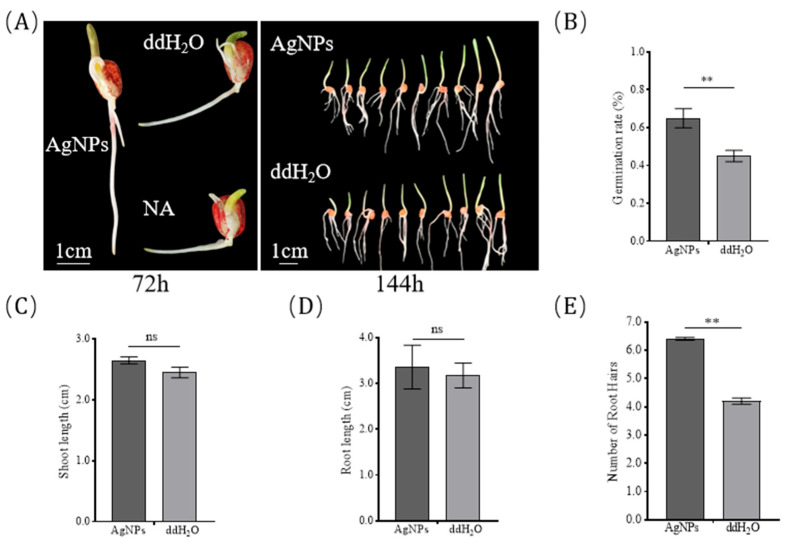
Phenotypic validation of AgNP stress tolerance in X178 inbred. Seedling phenotypes (**A**), germination rate (**B**), shoot length (**C**), root length (**D**), and number of lateral root (**E**) of X178 inbred grown for 4 days under non-treated and AgNP-treated conditions. Bar = 1 cm; *p* values calculated by one-way ANOVA; ** represents *p* < 0.01 (highly significant difference); ns represents no difference.

**Table 1 plants-14-03022-t001:** Transcriptome sequencing data statistics.

Lines	Rep	Total Reads	Rate of Total Mapped Reads (%)	Num. of Expressed Genes	Rate of Expressed Genes (%)
I178-CK	1	44,359,018	99.35	23,058	53.06
	2	43,820,934	99.37	23,675	54.48
I178-PT	1	44,696,386	98.73	22,826	52.52
	2	43,717,508	99.06	22,598	52.00
X178-CK	1	44,393,260	99.12	23,208	53.40
	2	45,159,698	99.04	22,980	52.88
X178-PT	1	43,138,354	98.87	23,723	54.59
	2	40,423,338	99.06	23,438	53.93

Note: Total reads: the number of reads after filtering the original data; rate of total mapped reads (%): the ratio of the remaining reads after filtering to the original unfiltered reads; num. of expressed genes: the total number of expressed genes with FPKM ≥ 1; rate of expressed genes (%): the proportion of the total number of expressed genes with FPKM ≥ 1 to the total number of genes.

## Data Availability

The data that support the findings of this study are available at www.ncbi.nlm.nih.gov/geo (accessed on 26 August 2025) with accession number PRJNA1309262.

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
