# Peer review of "Transcriptomic Analysis Reveals the Role of Silver Nanoparticles in Promoting Maize Germination"

_plants, 2025, doi:10.3390/plants14193022_

Round 1

Reviewer 1 Report

Comments and Suggestions for Authors

Dear authors, the manuscript is very well written. The data are interesting. However, I suggest improving the figures in terms of resolution and clarity of the captions. Please improve PCA descroption. I have left comments in the text.

Author Response

Dear reveviewer, we corrected all the questions and answered question by question as below attacged PDF file named "responses to the reviewers".

Reviewer 2 Report

Comments and Suggestions for Authors

- Line 23: Correct English grammar.

- Reference Citation: Please correct the reference citation format in the text to match the Plants journal's format.

- Figure 1: In the y-axis title, space the parentheses one space. Please review the other figures as well. Also, in (c), isn't it shoot length?

- Figure 2: In (b), it would be better to use the same color for each variety in the bar graph to visually enhance the visual distinction.

- Figure 3: The colors in (a) should be consistent with those in Figure 2, making it easier for readers to distinguish between the two. Furthermore, the "vs" in the x-axis should be applied consistently across all graphs, or removed altogether to ensure consistency.

- Figure 4: Include the meanings of BP, CC, and MF in the title to facilitate reader understanding.

- Lines 258-269: This section is an explanation of Figure 5 above. It would be better to reposition it elsewhere. Lines 258-269 --> In this section, please explain only the part about Figure 7.

- Figure 7: In the graph showing each parameter, is there a bar graph for NA? That means, it represents untreated seeds, not even H2O. Also, in this figure, please leave a space between parentheses. Also, isn't that shoot length?

- Figure 7: In Figure 1, when AG was applied to x178 seeds stored for one month, the germination rate was lower. However, in Figure 7, the AG treatment resulted in a higher germination rate. In other words, please provide an explanation so that readers can consistently predict the meaning of these results based on the figure and its title alone.

- Line 301: 'lots.Additionally,' --> lots. Additionally,

- In discussion: I would appreciate further refinement of the analysis of genes by qRT-PCR in Figure 5.

- Materials and Methods: Is there an explanation for NA, i.e., untreated seeds?

- Materials and Methods: Seeds were collected and stored at 95% humidity. I believe the humidity was excessively high. How was this humidity maintained?

- Line 410: During the germination rate,

- Lines 410-411: Ten seedlings were sampled for each replicate. So, how did you sample 40 seeds or seedlings here?

- I think more detailed explanations are needed for the Materials and Methods section.

- References: Please review and revise each section individually to ensure it conforms to the journal Plants' reference format.

Author Response

Please attached file for point-by point response.

Reviewer 3 Report

Comments and Suggestions for Authors
  • AgNP concentration (0.01 mM) and treatment duration (20 hours) lack proper justification. Include dose-response experiments or cite previous optimization studies.
  • The study claims I178 and X178 have "similar genetic background" but provides no genetic evidence. Include SNP analysis or pedigree information.
  • Figure quality is poor - graphs need better resolution and clearer labeling.
  • Some key results (e.g., specific gene functions) are buried in dense text.
  • Methods section needs more detail on AgNP characterization and preparation
  • Discussion is overly speculative in places; focus on data-supported conclusions
  • Reference formatting is inconsistent throughout

Author Response

please see below attached file for point-by-point response to the reviewers.

Round 2

Reviewer 2 Report

Comments and Suggestions for Authors

- Figure 7. : Which one is the shoot length between C and D? Also, the spelling "lenth" should be corrected to "length."

Author Response

-Comment: Figure 7. : Which one is the shoot length between C and D? Also, the spelling "lenth" should be corrected to "length."

-Answer: This is a writing error that has been corrected.

Reviewer 3 Report

Comments and Suggestions for Authors

The authors have addressed most of Reviewer's concerns, but several issues remain that prevent immediate acceptance. The study presents valuable findings on AgNP effects on maize germination, but methodological justifications and presentation quality still need improvement.

  • The authors claim to have "modified and described" the AgNP concentration (0.01 mM) and treatment duration (20 hours) in materials and methods, but the justification remains inadequate
  • While the authors reference previous work on genetic similarity between I178 and X178, the specific reference and evidence are not clearly presented
  • Some key findings in the result are still embedded in dense paragraphs

The manuscript presents novel and significant findings that warrant publication. However, the lack of proper justification for key experimental parameters and persistent presentation issues require minor revision before acceptance.

Author Response

Comment: The authors have addressed most of Reviewer's concerns, but several issues remain that prevent immediate acceptance. The study presents valuable findings on AgNP effects on maize germination, but methodological justifications and presentation quality still need improvement.

The authors claim to have "modified and described" the AgNP concentration (0.01 mM) and treatment duration (20 hours) in materials and methods, but the justification remains inadequate

While the authors reference previous work on genetic similarity between I178 and X178, the specific reference and evidence are not clearly presented

Some key findings in the result are still embedded in dense paragraphs

The manuscript presents novel and significant findings that warrant publication. However, the lack of proper justification for key experimental parameters and persistent presentation issues require minor revision before acceptance.

-Answer: There are very mature methods for the preparation of nano silver particles, and I have cited the most classical ones and included them in the references.

We have previously studied the genetic similarity between X178 and I178, so we did not repeat the description too much in the manuscript. This time, we added our previous study as a reference.

The questions raised in the conclusion and discussion section have been modified to make our research more clear.